# Changes in Bioactive Constituents in Black Rice Metabolites Under Different Processing Treatments

**DOI:** 10.3390/foods14091630

**Published:** 2025-05-05

**Authors:** Bin Hong, Shan Zhang, Di Yuan, Shan Shan, Jing-Yi Zhang, Di-Xin Sha, Da-Peng Chen, Wei-Wei Yin, Shu-Wen Lu, Chuan-Ying Ren

**Affiliations:** 1Food Processing Research Institute, Heilongjiang Academy of Agricultural Sciences, Harbin 150086, China; gru.hb@163.com (B.H.); zhangshanfood@163.com (S.Z.); yuandi199707@163.com (D.Y.); 18845896856@163.com (S.S.); 18846080235@139.com (J.-Y.Z.); shadixin1997@163.com (D.-X.S.); 2Heilongjiang Province Key Laboratory of Food Processing, Harbin 150086, China; 3Heilongjiang Province Engineering Research Center of Whole Grain Nutritious Food, Harbin 150086, China; 4Department of Food and Drug, Heilongjiang Vocational College of Agricultural Technology, Jiamusi 154007, China; chendapeng2003@126.com (D.-P.C.); yinweiwei5077@163.com (W.-W.Y.)

**Keywords:** black rice (BR), milling, germination, high temperature and pressure (HTP), metabolomics

## Abstract

In this study, liquid chromatography–mass spectrometry (LC-MS) was employed to conduct untargeted metabolomics analysis on black rice (BR), milled black rice (MBR), wet germinated black rice (WBR), and high-temperature and high-pressure-treated WBR (HTP-WBR). A total of 6988 positive ions and 7099 negative ions (multiple difference ≥1.2 or ≤0.8333, *p* < 0.05, and variable importance in projection ≥1) were isolated, and 98 and 100 differential metabolic pathways were identified between the different samples in the positive and negative ion modes, respectively. Distinctive variations in the metabolic compositions of BR, MBR, WBR, and HTP-WBR were observed. Flavonoids, fatty acids, lipids, phenylpropanoids, polyketides, benzenoids, and organooxygen were the dominant differential metabolites. Milling removed the majority of bran-associated bioactive components such as phenolic acids, anthocyanins, micronutrients, fatty acids, antioxidants, and dietary fiber. The germination process significantly reduced the number of flavonoids, polyketides, and lipid-related metabolites, while enzymatic activation notably increased the number of organic acids and amino acids. HTP treatment synergistically enhanced the content of heat-stable flavonoids and polyketides, while simultaneously promoting fatty acid β-oxidation. These findings establish novel theoretical foundations for optimizing processing methodologies and advancing functional characterization in black rice product development.

## 1. Introduction

Black rice (BR) is rich in protein, fat, minerals, dietary fiber, and other common nutrients. Compared to other rice varieties, black rice is distinguished by its high content of anthocyanins, flavonoids, phenolic acids, γ-aminobutyric acid, and essential minerals (e.g., manganese, zinc, and copper) [1,2]. These nutrients in BR grains have antioxidant, anti-aging, and anti-inflammatory effects, reduce blood lipid levels, treat diabetes, and protect the nervous system and retina [3]. Therefore, research on BR has been attracting increasing attention from many researchers and has made remarkable progress. In recent years, owing to the development of the economy and interest in healthy lifestyles, people’s demand for BR products with high nutritional values has been increasing. BR originates from a functional mutation of the OsB2 locus. OsB2 is a bHLH gene that regulates anthocyanidin synthesis in rice. Altering the promoter region of OsB2015 leads to a significant enhancement in anthocyanidin expression in both the seeds and leaves [4]. The main types of anthocyanins in black rice are cyanidin and peonidin. These two anthocyanins are primarily responsible for the deep purple or black color of black rice and also endow it with rich antioxidant properties. Cyanidin is the most predominant anthocyanin in black rice, while peonidin is typically present in lower concentrations. These anthocyanins not only contribute to the color of black rice, but also offer potential health benefits, such as anti-inflammatory, antioxidant, and cardiovascular protective effects. BR contains significantly more Arg, Asn, Glu, Met, and Orn than other colors of rice; this proves that BR has a higher nutritional value [5]. The nutrients in BR are mainly concentrated in the bran layer, and the total concentration of phenolics, flavonoids, and anthocyanins is significantly reduced from the outermost to the innermost layer [6]. The amount and effectiveness of the nutrients in BR are closely related to not only the synthesis of substances in the plants during the early stages, but also the storage, milling, cooking, and germination of rice after harvest [7]. Milling is performed to destroy the waxy layer and improve the cooking quality of grains; however, it results in the loss of nutrients [8]. Bioactive phytochemicals are not uniformly distributed in rice grains, and the higher the degree of milling, the fewer phytochemicals in milled rice [9]. The process of germination leads to better sensory quality and higher yields of bioactive substances in brown rice. The chemical composition significantly changes during the germination of brown rice [10]. Brown rice germination leads to increased amounts of essential nutrients, such as potassium, zinc, magnesium, vitamins E, B1, and B6, lysine, γ-aminobutyric acid (GABA), and γ-oryzanol [11]. Moreover, anti-nutritional factors break down during the germination process due to the activation of intrinsic enzymes such as α-amylase, pullulanase, phytase, and various glucosidases. During the germination of rice grains, α-amylase is activated and the increased enzyme activity promotes the breakdown of anti-nutritional factors, while other enzymes in the germination process (such as phytase) exhibit synergistic action to degrade phytic acid. Furthermore, additional hydrolytic enzymes may be activated during germination, collectively reducing the content of anti-nutritional factors [12]. The concentrations of total phenolics, flavonoids, and tannins in germinated rice are significantly greater compared to those in non-germinated rice [13]. Germinated brown rice has a softer texture than native brown rice [10]. The germination of BR can not only significantly increase the concentrations of GABA, gluin, and other nutrients, but also lead to the high nutritional value of germination products owing to the large number of bioactive substances [14]. Heat–moisture treatment is regarded as a safe, physical modification method suitable for human consumption. Heat–moisture treatment can improve the physicochemical properties of germinated rice [15]. In recent years, heat–moisture treatment has been widely used in germinated brown rice and applied to rice-based products, such as noodles and cookies [16,17]. High-temperature and high-pressure treatments are some of the most common sterilization methods in the food industry. For germinated black rice products, these treatments not only extend shelf life, but they also induce metabolic changes in their nutritional components.

Current studies on black rice primarily focus on profiling its nutritional and bioactive components (e.g., anthocyanins and phenolic acids), whereas limited evidence exists regarding how these compounds modulate human metabolic pathways or their efficacy in improving metabolic health through clinical interventions. Investigating the metabolic regulatory effects of black rice products on humans is critical for elucidating their health benefits and advancing the development of functional foods derived from black rice. Metabolomics is used not only to comprehensively analyze the changes in metabolite concentrations, but also to clarify the change rules between different metabolites and various treatment conditions to evaluate the biological reactions and effects of different substances [18]. Although metabolomics has been widely applied in food science, its utilization in elucidating metabolic changes during black rice processing—particularly under combined germination and high-temperature, high-pressure treatments—remains underexplored. In this study, liquid chromatography–tandem mass spectrometry (LC-MS/MS) was performed to examine all metabolites in BR, BR after milling (MBR), wet germinated black rice (WBR), and high-temperature- and high-pressure-treated germinated BR (HTP-WBR). Multiple statistical analysis methods, including principal component, cluster, and pathway analyses, were used to analyze the changes in the metabolite species and quantity between BR, MBR, WBR, and HTP-WBR and ascertain the key metabolic pathways involved in metabolite changes. This study bridges the critical research gap in the metabolite alterations of black rice products under diverse processing methods, offering novel insights into their functional and mechanistic responses to milling, germination and high-temperature and high-pressure treatments. The results provide a new basis for milling, germination, and HTP technology and the functional evaluation of BR.

## 2. Materials and Methods

### 2.1. Materials and Instruments

#### 2.1.1. Instruments

A QH-62 germination cabinet (Qingzhou Qinghua Bean Sprout Machinery Equipment Co., Ltd., Qingzhou, China) was used.

A Yamamoto VP-32 Experimental Rice Milling Machine (Satake Corporation, Hiroshima, Japan) was used.

A BXM-60EI Vertical Pressure Steam Sterilizer (Shanghai Boxun Medical Biological Instrument Co., Ltd., Shanghai, China) was used.

A liquid chromatograph (2777C UPLC system, Waters, UK) and a mass spectrometer (Xevo G2-XS QTOF, Waters, UK) were used for LC-MS.

#### 2.1.2. Materials

Black Pearl rice was planted in Weiguo Township, Wuchang City, Heilongjiang Province.

### 2.2. Methods

#### 2.2.1. Different Treatments of BR Samples

BR was peeled and processed into BR with no shell and ground into rice after milling (bran-free layer and embryo; MBR). After washing and soaking, the BR was germinated in a germination cabinet at 30 °C for 42 h, with a spray interval of 1 h, to produce WBR. The produced WBR exhibited a moisture content of 35–38% (wet basis) and was subsequently processed at 115 °C and for 20 min within a pressure sterilization vessel (HTP-WBR). Approximately 5 g of whole grain rice from each sample was weighed and placed into 10 mL centrifuge tubes, which were then sealed. This procedure was repeated in triplicate for each sample to ensure accuracy. Following this, the samples were rapidly frozen using liquid nitrogen and stored at −80 °C in a freezer, awaiting further analysis.

#### 2.2.2. Metabolite Extraction

The test samples stored at −80 °C were retrieved, and one grain of rice (with a single-grain mass of 21–23 mg) was immediately selected from each tube. The extraction process was carried out by proportionally adding reagents based on the actual mass of each grain (measured to the nearest 0.0001 g) to achieve the following 25 mg system for metabolite extraction. The specific procedure was as follows: a total of 25 mg of the test sample was transferred into a 1.5 mL EP tube, followed by the addition of 800 µL of a pre-chilled precipitation agent (methanol–acetonitrile–water, 2:2:1, *V*/*V*). Two small steel beads were added, and the mixture was ground in a grinder (60 Hz, 4 min). After removing the steel beads, the sample was subjected to ice bath sonication (80 Hz, 10 min) and then placed in a −2 °C freezer for 120 min. Subsequently, the sample was centrifuged (4 °C, 25,000× *g*, 15 min), and 600 µL of the supernatant was collected. The supernatant was transferred to a freeze-dryer for lyophilization. After freeze-drying, the sample was reconstituted with 600 µL of a 10% methanol solution, followed by ice bath sonication (80 Hz, 10 min) and centrifugation (25,000× *g*, 4 °C, 15 min). The supernatant was collected, and 50 µL was taken from each of the six samples to create a pooled sample (quality control, QC).

#### 2.2.3. Liquid Chromatography and Tandem MS Conditions

ACQUITY UPLC HSS T3 column (100 × 2.1 mm, 1.8 μm) was used at 50 °C with a flow rate of 0.4 mL/min. Mobile phases: A, 0.1% formic acid; B, 0.1% formic acid in methanol. Gradient: 0–2 min, 100% A; 2–11 min, 100% to 0% A, 0% to 100% B; 11–13 min, 100% B; and 13–15 min, return to 100% A. Injection volume: 5 µL per sample.

The Xevo G2-XS QTOF high-resolution tandem mass spectrometer was used to analyze small molecules eluted from the LC column in both positive and negative ion modes. Positive ion mode: cone voltage 40.0 V, capillary voltage 3.0 kV; negative ion mode: cone voltage 40.0 V, capillary voltage 2.0 kV. Data were acquired in full-scan MSE mode: primary scan (50–1200 Da, 0.2 s), precursor ion fragmentation (20–40 eV), and secondary scan (0.2 s) for fragment ion collection. Real-time mass correction was performed every 3 s using the LE signal. A QC sample was analyzed before test samples to monitor instrument stability.

#### 2.2.4. Metabolomics Analysis Based on Advanced LC-MS

In this study, an advanced mass spectrometer Xevo G2-XS QTOF (Waters, UK) was used for the data acquisition of the rice samples (BR, MBR, WBR, and HTP-WBR), commercial software Progenesis QI (version 2.2) (Waters, UK), and the metabolomics software package MetaX was used for the mass spectrometry data analysis, wherein identification was based on the HMDB and KEGG and LipidMaps databases. The project uses the fold change (FC) and *p*-value of univariate analysis to choose differentially expressed metabolites. The filtering rules are as follows: (1) fold change ≥ 1.2 or ≤0.8333; (2) *p*-value < 0.05, and all must be met for an ion to be considered as a differential ion. The data preprocessing was performed using the MetaX software, which included filling in missing values using the KNN method for the extracted data, removing low-quality ions (those with more than 50% missing values in QC samples or more than 80% missing values in actual samples), followed by filtering the data to exclude ions with a relative standard deviation (RSD) > 30% in all QC samples (ions with RSD > 30% exhibit significant fluctuations during the experiment are not included in downstream statistical analysis). Finally, the statistical information on the remaining ions was obtained. Batch effects were corrected through a QC-based robust LOESS signal correction approach. Furthermore, Principal Component Analysis (PCA) was carried out with the SIMCA software (version 14.1).

#### 2.2.5. Peak Extraction and Identification

The entire process was primarily conducted using the software Progenesis QI (version 2.2), encompassing steps such as peak alignment, peak extraction, normalization, spectral interpretation, and compound identification. The key parameter settings for peak extraction and identification are detailed in Table 1.

#### 2.2.6. Identification of the Differential Metabolites

Statistical analysis was performed using the self-developed metabolomic MetaX software package and Progenesis QI (ver. 2.2), and the metabolites were identified from the Kyoto Encyclopedia of Genes and Genomes (KEGG) and LipidMaps databases and the Human Metabolome Database (HMDB). Multivariate analysis was performed on the variable importance in projection (VIP) scores of the first two principal components from the partial least squares discriminant analysis (PLS-DA) model, combined with *p*-values and univariate analysis for multi-difference screening. The screening conditions were as follows: (1) VIP ≥ 1, (2) *p* < 0.053, and (3) multiple difference ≤ 0.8333 or ≥1.2. The three conditions were crossed, and the common metabolites were the differential metabolites. The final results were presented as the VIP values, *p*-values, and multiple differences.

#### 2.2.7. Differential Metabolite Pathway Analysis

Metabolic pathway analysis can help to understand the major biochemical metabolic pathways and signal transduction pathways involved in metabolites. This study is based on the KEGG database for the annotation of metabolites’ metabolic pathways.

#### 2.2.8. Data Processing

First, peak list information was extracted from the raw data obtained through mass spectrometry separation and identification. The data underwent preprocessing and correction, followed by statistical analysis, as described in Section 2.2.5, to identify differentially expressed metabolites between samples. Subsequently, the metabolites and differential metabolites were annotated by referencing the HMDB and the LipidMaps and KEGG databases. Pathway analysis was performed using the KEGG pathway database to determine the primary metabolic pathways associated with the differential metabolites.

Univariate analysis included non-parametric or parametric tests and fold change analysis. Multivariate analysis encompassed Principal Component Analysis (PCA) and cluster analysis.

## 3. Results

### 3.1. Number of Metabolites

An untargeted metabolomic assay was performed on the BR, MBR, GBR, and HTP-WBR samples (based on the KEGG database, HMDB, and LipidMaps analysis). In Table 2, a total of 8140 positive ions and 9940 negative ions were identified. From these, 6988 positive ions and 7099 negative ions with a relative standard deviation (RSD) of ≤30% were selected for further statistical analysis. Initial identification yielded 5011 positive ions and 4923 negative ions, while secondary identification resulted in 3026 positive ions and 3452 negative ions. Critically, these ions correspond to metabolites spanning key functional classes, including the following: lipids (e.g., glycerophospholipids and sphingolipids), which influence membrane stability and bioactive compound delivery in processed grains; phenolic acids and flavonoids (e.g., anthocyanin derivatives), known for their antioxidant and anti-inflammatory roles in human health; amino acid derivatives (e.g., γ-aminobutyric acid and GABA), linked to neuroprotective effects and stress response modulation in germinated grains; and organic acids (e.g., citric acid), which play a critical role in improving nutrient bioavailability and reinforcing antioxidant potential in BR-derived processed foods.

### 3.2. Classification and Functional Analysis of the Metabolites

The identified ions were classified and functionally annotated based on the KEGG database and HMDB (univariate analysis of fold change and *p*-value values were used to screen differentially expressed metabolites. Screening conditions: (1) fold change ≥ 1.2 or ≤ 0.8333; (2) *p*-value < 0.05, the three take the intersection to obtain the shared ion, which is the differential ion.), and organized into three major categories: compounds having biological functions, lipids, and phytochemical compounds. In the positive ion mode, 69,650 first-order metabolites were identified. In the negative ion mode, 45,771 first-order metabolites were identified. Among the first-order metabolites, fatty acids, lipids, phenylpropanoids, polyketides, benzenoids, and organic compounds were the main metabolites. Flavonoids, terpenoids, amino acids, polypeptides, and carbohydrates were the main components of the secondary metabolites. The metabolite quantities are shown in Figure 1.

The identified metabolites and their annotated pathways play critical roles in maintaining biological homeostasis and conferring health benefits through multiple mechanisms: 1. Primary metabolite contributions: Fatty acids and lipids—serve as structural components of cell membranes, energy reservoirs, and precursors for signaling molecules that regulate inflammation and cardiovascular function; phenylpropanoids and benzenoids—exhibit potent antioxidant activities through free radical scavenging, protecting against oxidative stress-related diseases including cancer and neurodegeneration; and polyketides—many demonstrate antimicrobial and antitumor properties through the targeted inhibition of pathogenic enzymes. 2. Secondary metabolite functions: flavonoids—modulate inflammatory pathways and improve vascular endothelial function via NO signaling; terpenoids—enhance immune responses through MHC molecule regulation and demonstrate neuroprotective effects via Nrf2 pathway activation; and amino acid derivatives—regulate neurotransmitter synthesis and participate in detoxification processes through urea cycle modulation.

According to the KEGG pathways enriched in positive ions and based on the functional roles of metabolites in biological systems, 103, 963, and 1297 metabolites were annotated for biological information processing, biochemical transformations, and nutrient metabolism, respectively. In the case of negative ions, 122, 860, and 1351 metabolites were annotated for biological information processing, biochemical transformations, and nutrient metabolism, respectively.

### 3.3. Differences in the Classification and Quantity of Metabolites in Black Rice Under Various Processing Methods

The trend of separation, and whether there are abnormal points reflecting the variability between rice samples, was first observed by performing a PCA assay. In Figure 2, BR, MBR, WBR, and HTP-WBR are separated well in the PCA models for both positive and negative ions. The principal component of BR is significantly different from those of the other three samples, and the principal component of WBR is relatively close to that of HTP-WBR. The significant separation of BR from other samples indicates that unprocessed black rice is rich in unique primary and secondary metabolites. The aleurone layer and seed coat of black rice contain high concentrations of polyphenols and anthocyanins, which are potent antioxidants significantly reduced during milling (MBR) due to the removal of these outer layers. BR contains higher levels of long-chain unsaturated fatty acids, associated with its intact endosperm structure, while oxidative degradation during processing leads to increased short-chain aldehyde and ketone metabolites in MBR. The metabolic profiles of WBR (germinated black rice) and HTP-WBR are closely aligned, demonstrating that the germination process dominates metabolic reprogramming, and that the high-temperature and pressurized (HTP) treatment does not completely mask germination-induced changes. Although some heat-sensitive metabolites (e.g., vitamin B1 and certain flavonoid glycosides) decrease in HTP-WBR, Maillard reaction products (e.g., melanoidins and furans) promoted by high temperatures form a complementary relationship with germination-derived metabolites (e.g., GABA and ferulic acid), maintaining the overall similarity of their metabolic profiles. The distinct separation of MBR from other samples confirms the substantial loss of functional metabolites caused by the removal of the bran layer during milling.

The heatmap illustrating the cluster analysis of all differential metabolites among the four black rice samples is presented in Figure 3. The varying colors represent different concentration levels: red indicates upregulation, green indicates downregulation, and the gradient from green to red signifies an increase in the expression level of the metabolites from low to high. It can be observed that the same sample exhibits similar characteristic expressions in identical regions, indicating a high degree of similarity within the same sample. Conversely, different samples show distinct characteristic expressions in the same regions, highlighting significant differences between the samples.

The numbers of differential ions in the different forms of BR are listed in Table 3. In both the positive and negative ion modes, the number of differential ions in BR:MBR is significantly higher than those in the other two groups. The primary identification results reveal 3075 differential positive ions (1698 upregulated and 1377 downregulated) and 4434 differential negative ions (2245 upregulated and 2189 downregulated) between BR and MBR. Polyketides, fatty acids, lipids, benzenoids, and substituted derivatives are the most significant differential metabolites in both the positive and negative ion modes. A total of 3739 differential positive ions (1420 upregulated and 2319 downregulated) and 5267 differential negative ions (2191 upregulated and 3076 downregulated) are revealed between BR and MBR. Flavonoids, organooxygen compounds, carbohydrates, and carboxylic acids are the most significant differential metabolites in both the positive and negative ion modes. A similar result is obtained for BR:WBR and WBR:HTP-WBR.

The top seven classifications for the numbers of differential metabolites between different BR forms are shown in Figure 4. After being milled (BR:MBR), most BR metabolites are downregulated. Similarly, after germination (BR:WBR), most metabolites are downregulated. Flavonoids, polyketides, fatty acids, and other metabolites are significantly different. After high-temperature and high-pressure treatment (WBR:HTP-WBR), fatty acids and flavonoids are significantly downregulated in both modes.

### 3.4. Differential Metabolic Pathways in Black Rice Under Various Processing Methods

The primary biochemical pathways associated with the differential metabolites were enriched and analyzed using the KEGG database. In the positive and negative ion modes, 98 and 100 differential pathways were identified between the different samples, respectively. As shown in Figure 5, the primary metabolic pathways were involved in more than 40 metabolites. According to the number of metabolites in the positive and negative ion modes, the metabolic pathways differ significantly, mainly in the biosynthesis of terpenoids, flavonoids, isoquinoline alkaloids, and amino acids, and in the metabolism of 2-carboxylic acid, arachidonic acid, and tyrosine.

The main metabolites found in the four BR samples are fatty acids, lipids, lipid-like molecules, flavonoids, phenylpropanoids, and polyketides. The metabolic pathways involved are flavonoid, amino acid, and terpenoid biosyntheses and arachidonic acid metabolism. The metabolic pathways involved in the primary differential metabolites in each sample are listed in Table 4.

Different processing methods significantly impact the composition and health functionalities of black rice metabolites; MBR causes the substantial loss of outer-layer flavonoids (e.g., anthocyanins) and γ-oryzanol, while inducing the accumulation of oxidative byproducts, necessitating nutritional fortification. Germination (WBR) partially reduces flavonoids, but activates enzymatic reactions to significantly enhance GABA and energy-related metabolites, conferring antioxidant, neuroregulatory, and exercise endurance-boosting properties, making it suitable for functional food development. HTP-WBR degrades heat-sensitive components (e.g., flavonoid glycosides and amino acids) but generates melanoidins via the Maillard reaction and suppresses pro-inflammatory factors, offering dual benefits in flavor enhancement and anti-inflammatory potential. Processed black rice derivatives can be strategically applied as natural antioxidants (WBR), cardiovascular healthy oils (BR), anti-inflammatory medical foods (HTP-WBR), and natural preservatives, reflecting a metabolomics-guided precision nutrition development strategy.

## 4. Discussion

Brown rice has become increasingly popular thanks to its high nutritional quality, which is recognized worldwide. Brown rice is richer in fibers, iron, calcium, vitamins, and minerals than polished rice [19]. BR contains many bioactive ingredients, such as anthocyanins, phenolic compounds, gluins, phytosterols, and phytic acid. Because of its potential nutritional value, BR has been increasingly favored by consumers and researchers in recent years [20]. An LC-MS-based untargeted metabolomic analysis was conducted in this study to uncover differential metabolites. By analyzing the non-targeted metabolites of BR and its various processed samples (MBR, WBR, and HTP-WBR), we concluded that the number of differential metabolites between the various samples is extremely high. The types of metabolites are mainly fatty acids, lipids, flavonoids, phenylpropanoids, polyketones, and benzenones. BR contains high levels of anthocyanins, carotenoids, phytosterols, tocopherols, protocatechuic acid, and other phenolics [21,22]. It also contains high levels of metabolites originating from the terpenoid and phenylpropanoid biosynthesis pathways [23]. PCA assay showed that BR, MBR, WBR, and HTP-WBR were separated in both modes. As PCA effectively represents the accumulation patterns and metabolite specificity within the samples, this result indicates the presence of metabolomic differences between BR, MBR, WBR, and HTP-WBR.

As shown in the volcano plots of BR and MBR in Figure 6a,b, the number of differential metabolites decreases significantly after BR is milled. As shown in Figure 4a,b, among the differential metabolites of BR and MBR, fatty acid and lipid quantities change slightly, and the main changes are concentrated in flavonoids, polyketides, and organic acid compounds. Metabolic pathway analysis of the differential metabolites in BR:MBR reveals that the metabolic pathways of flavonoid biosynthesis and arachidonic acid metabolism are the most significantly upregulated in the positive and negative ions. The major differential metabolites involved in these two metabolic pathways are phenylpropanoids, polyketides, and organic acids (Table 3). Milling is an important factor in rice-processing technology. Moderate grinding helps to improve the taste and appearance of rice. However, milling leads to a continuous reduction in the concentrations of active substances such as phenolic acids and anthocyanins in BR, and the degree of reduction is correlated with the degree of milling [24]. The milling process discards most micronutrients, fatty acids, antioxidants, and fibers. As a result, diets that are overly reliant on white rice can cause deficiencies in several nutritional factors [25]. Black rice has a significant advantage over brown rice due to its higher content of anthocyanins [26]. However, a comparative analysis of Figure 4a,b, clearly shows that the content of flavonoids significantly decreases after milling treatment. Anthocyanins are severely reduced during the milling process of black rice, which greatly affects its nutritional value. More than 10 species of anthocyanins have been found in BR, and these anthocyanins are mainly distributed in the skin vacuoles of BR species. Few colorless anthocyanins are present in the aleurone cell wall, with little or no presence in the embryo and endosperm [27]. In cereal grains, the pericarp produces anthocyanins, whereas the endosperm lacks anthocyanins [28,29]. The phenolic acid concentration continuously decreases with an increase in the grinding degree [30]. As milling proceeds, the contact surface of free phenolic acid with oxygen increases, and the oxidation rate of phenolic acid increases [31].

Phenolic acids, anthocyanins, and other bioactive compounds are predominantly distributed in the outer layers of black rice. During the milling process, the gradual removal of these outer tissues naturally reduces the content of these bioactive substances. Additionally, for phenolic acids, increased exposure to oxygen during milling accelerates their oxidation rate. The phenolic hydroxyl groups in phenolic acids are readily oxidized by oxygen, forming quinones and other oxidation products, leading to a decline in their content. For anthocyanins, mechanical forces during milling may disrupt their chemical structures, destabilizing them and causing significant degradation.

The milling process discards most micronutrients, fatty acids, antioxidants, and dietary fiber. Deficiencies in certain vitamins (e.g., B vitamins) may impair energy metabolism and neurological functions, while mineral losses (e.g., iron and zinc) could contribute to anemia or delayed growth. The substantial loss of fatty acids during milling may compromise cell membrane fluidity and functionality, thereby disrupting normal cellular physiological activities.

In Figure 6c,d, the volcano plots of WBR:BR are significantly more downregulated than upregulated. The amounts of the differential metabolites decrease significantly after BR germination. As shown in Figure 4c,d, the numbers of flavonoids, polyketides, fatty acids, and lipid metabolites decrease significantly. The concentrations of organic acids and amino acids increase significantly during germination. The metabolic pathways of flavonoid biosynthesis and arachidonic acid metabolism are the most divergent metabolic pathways after BR germination (Table 3). The metabolites that pass through these two metabolic pathways are mainly fatty acids, lipids, polyketides, and organic acids. Germination, which includes both preharvest sprouting and controlled germination, is a biological process initiated by the activation of endogenous enzymes [11]. The germination process leads to an increase in the levels of particular bio-functional components, such as GABA, lysine, B-group vitamins, and several antioxidants (γ-oryzanol, vitamin E, and phenolic compounds) [32,33]. Following the germination process, WBR exhibited a notable 68% elevation in GABA content relative to BR [14]. This marked increase in the GABA concentration suggests the partial proteolysis of storage proteins and their subsequent translocation to the actively growing regions of the grains. Concurrently, the enzymatic activity of glutamate decarboxylase is upregulated, catalyzing the conversion of glutamic acid to GABA [34]. Thanks to the spraying germination method, a large amount of water is sprayed on BR during the germination process, which causes a considerable loss of flavonoids and polyketides from BR skin. This is consistent with the conclusions of a previous study [35]. Anthocyanins are water-soluble compounds found in the pericarp of rice, and their concentration decreases with germination [36]. Future research on germinated black rice should prioritize the optimization of germination techniques to enhance the production of bioactive compounds while simultaneously maximizing the retention of anthocyanins. This dual-focused approach aims to unlock the full nutritional and functional potential of black rice through controlled germination processes. The decline in the amounts of fatty acid and lipid metabolites may be due to lipolysis, which occurs during the first stages of grain germination, uses fat as a source of energy, and leads to changes in fat [37]. Triacylglycerol is hydrolyzed by enzymes during germination to release free fatty acids, which undergo β-oxidation to generate essential energy for seed growth. Therefore, a decrease in the crude lipid concentration is expected during germination [38]. Ash and many minerals are lost during germination, possibly because they are required as coenzymes for the catalysis of proteins and carbohydrates that are transferred to the radicle [39,40]. The total free amino acid concentration increases with an increasing germination time [41]. During the germination process, endogenous proteases are activated, breaking down prolamin and glutelin in the endosperm into peptides and amino acids, which serve as nitrogen sources for seed germination. Meanwhile, the newly generated amino acids are transported to the germ, where they are utilized as nutrients, and the grain cells undergo translation and transcription to synthesize new proteins [42]. Free phenolic compound amounts generally increase during germination because enzymatic activity promotes the breakdown of interactions between phenolic compounds and macromolecules [43]. Germination led to a significant 63.2% increase in the total phenolic content. This rise can be attributed to the growth of cells during the germination process, which likely enhances the levels of most bound phenolics [44].

The numbers of flavonoids, polyketides, fatty acids, and lipid metabolites significantly decrease during the germination process. For flavonoids and polyketides, the sprinkling germination method leads to their substantial leaching from the bran layer. Additionally, the biosynthesis of flavonoids and polyketides, typically regulated by a series of enzymes, may be suppressed due to altered metabolic pathways during germination that inhibit enzyme activity. The reduction in fatty acids and lipid metabolites is likely caused by lipolysis. In the early stages of grain germination, stored fats are utilized as an energy source: triacylglycerols are enzymatically hydrolyzed into free fatty acids, which subsequently undergo β-oxidation to generate energy. Lipase activity is activated during germination, accelerating triacylglycerol hydrolysis. Concurrently, enzymes in the β-oxidation pathway exhibit enhanced activity under germination conditions, rapidly consuming free fatty acids and reducing their overall content. In contrast, organic acids and amino acids show significant increases in concentration during germination. For organic acids, alterations in plant respiration pathways during germination lead to the accumulation of intermediates in the TCA cycle (e.g., citric acid and malic acid). Furthermore, specific organic acid synthesis pathways, such as the glyoxylate cycle, are activated during early germination. This cycle converts stored fats into carbohydrates, producing organic acids like malic acid. For amino acids, endogenous proteases are activated during germination, breaking down storage proteins (e.g., prolamins and glutelins) in the endosperm into peptides and amino acids. A portion of the newly generated amino acids is transported to the embryo to synthesize proteins essential for seedling growth, while the remainder accumulates intracellularly, increasing free amino acid concentrations. Prolonged germination further enhances protease activity, continuously elevating amino acid levels. Post germination, black rice exhibits significantly elevated levels of GABA and antioxidants (γ-oryzanol, vitamin E, and phenolic compounds). After ingestion, GABA is absorbed through the intestines into the bloodstream and crosses the blood–brain barrier to exert neuromodulatory effects. The increased antioxidants enhance the body’s antioxidant defense system, reducing oxidative stress and promoting overall health.

In Figure 6e,f, the volcano plots of HTP-WBR:WBR show no significant differences in the upregulation and downregulation of the differential metabolites. According to Figure 4e,f, the flavonoid and polyketide concentrations are significantly elevated, and the fatty acid concentration decreases significantly after heat and pressure treatment. WBR after heat treatment shows a 9% increase in anthocyanin concentration compared with the control WBR; similar findings were reported in previous studies [14,45]. Heat treatment after germination increases the GABA levels because germination probably promotes water absorption and provides a high-moisture environment to activate the relevant endogenous enzymes, which trigger the synthesis of GABA [46]. Heat treatment during cooking methods significantly increases the total amount of free phenolic acids [42]. Thermal processing induced the degradation of poly-phenolic compounds, with anthocyanins being particularly susceptible to thermal decomposition. Simultaneously, heat treatment facilitated the formation of hydrogen bonding interactions between phenolic compounds, vitamins, and starch molecules [47]. The significant decrease may be attributed to the activation of enzymes during bioprocessing, as well as process-induced alterations in the availability of nutrients and bioactive components [48]. The hydrothermal conditions could potentially activate key enzymes involved in both the L-phenylalanine and L-tyrosine metabolic pathways, thereby enhancing phenolic acid biosynthesis [49]. Furthermore, anthocyanins may undergo degradation, initially converting to anthocyanin aglycones, which subsequently decompose into phenolic acids, ultimately leading to an accumulation of phenolic acid content. The duration of cooking exhibited minimal impact on metabolite alterations, whereas the application of pressurized conditions induced substantial modifications in black rice metabolites compared to atmospheric pressure treatment. This observation aligns with the findings reported by Peres et al. in their investigation of cooking duration and pressure effects on phytochemical profiles in wild rice [50]. The results indicate that heat treatment reduces the fat amount, and that the fat concentration in food decreases with an increasing heating time [51]. Generally, processing food using heat causes fat damage, and the level of damage is highly dependent on the temperature and duration of heating. The heating process melts fat, turns fat into free fatty acids, or even evaporates fat to lead to other aspects such as flavor [2].

After high-temperature and high-pressure treatment, the concentrations of flavonoids and polyketides increase markedly. This is likely due to the activation of the relevant enzymes via thermal pressure conditions, which upregulates the activities or gene expressions of key enzymes like phenylalanine ammonia lyase (PAL) and chalcone synthase (CHS). For polyketides, the intracellular microenvironment is altered, enhancing the activity of polyketide synthase (PKS) and promoting synthesis. Flavonoids possess antioxidant and anti-inflammatory properties, capable of scavenging free radicals and reducing the risk of chronic diseases. Polyketides exhibit diverse biological activities; some have antibacterial functions and can modulate the immune system. The fatty acid concentration decreases significantly post high-temperature and high-pressure treatment. Firstly, a high temperature promotes fatty acid oxidation, resulting in chain breakage and the formation of oxidation products. Secondly, thermal pressure treatment activates lipase, accelerating fat hydrolysis and leading to fatty acid decomposition or participation in other metabolic pathways. Unsaturated fatty acids are beneficial for cardiovascular health. The decline in fatty acid concentration caused by thermal pressure treatment impacts the fatty acid nutritional composition in food and human intake, thus weakening the health benefits. Thermal processing degrades polyphenolic compounds, including anthocyanins, which are heat-sensitive. As the temperature rises and processing time extends, anthocyanins decompose into small-molecule phenolic acids. However, within a specific temperature range, heat treatment can convert anthocyanins from the bound to the free state, increasing their content. High-concentration anthocyanins enhance foods’ antioxidant capacity, benefiting health. But excessive heating causing substantial anthocyanin degradation diminishes antioxidant functions. Heat treatment after germination elevates γ-aminobutyric acid (GABA) levels. During germination, seeds absorb water, creating a high-humidity environment that activates glutamate decarboxylase (GAD), catalyzing glutamate decarboxylation to produce GABA. GABA regulates nervous system excitability, alleviating anxiety and improving sleep quality.

## 5. Conclusions

In this study, a UPLC-MS/MS-based metabolomics technique was used to identify different metabolites in different processed forms of black rice. Differences in metabolic composition were observed among BR, MBR, WBR, and HTP-WBR. Flavonoids, fatty acids, lipids, phenylpropanoids, polyketides, benzenoids, and organooxygen were the dominant differential metabolites. The differences between the four samples were analyzed by examining different metabolites and metabolic pathways in BR vs. MBR, BR vs. WBR, and WBR vs. HTP-WBR. Through differential metabolite analysis, when compared with brown rice (BR), the milling of BR resulted in the removal of more than 80% of bran-associated phenolic acids, anthocyanins, micronutrients, fatty acids, antioxidants, and dietary fiber. The germination process brought about significant decreases in flavonoids, polyketides, fatty acids, and lipid metabolites. Conversely, organic acids and amino acids showed remarkable increases during germination. High-temperature and high-pressure (HTP) treatments synergistically enhanced the content of heat-stable flavonoids and polyketides and accelerated the metabolism of fatty acids. The germination of brown rice can increase the content of bioactive substances. However, due to the dissolution and loss of water-soluble substances during the germination process, the contents of anthocyanins and phenolic acids in the germinated black rice products decrease significantly. Subsequent research could be concentrated on the methods of maintaining the nutrients during the germination of colored rice varieties.

Further research is needed to validate the differential metabolites in specific rice forms, particularly in WBR and HTP-WBR. We plan to employ a targeted metabolomics approach to quantify key metabolites and analyze the changes in critical metabolites in black rice.

## Figures and Tables

**Figure 1 foods-14-01630-f001:**
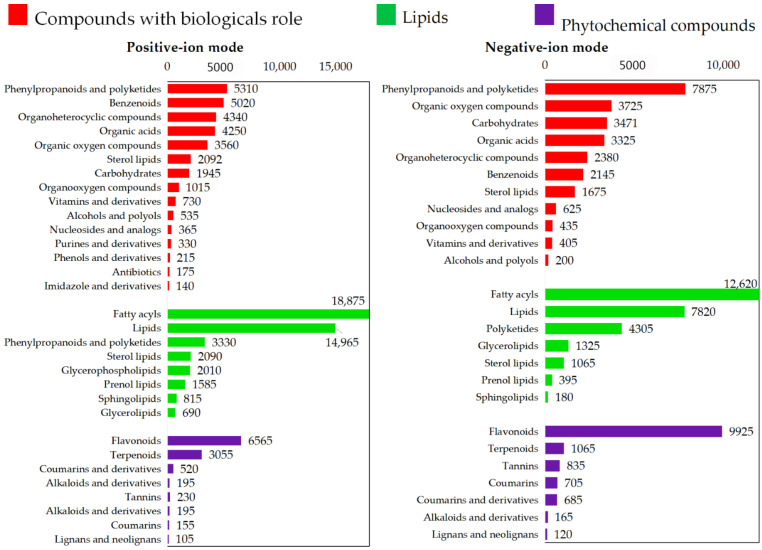
Classification of identified metabolites as per the KEGG database and HMDB.

**Figure 2 foods-14-01630-f002:**
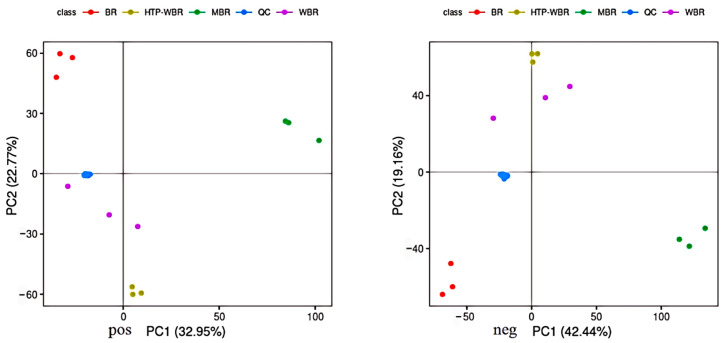
PCA assay of BR, MBR, WBR, and HTP-WBR. Each point in the figure corresponds to a sample, and the varying colors denote different groups.

**Figure 3 foods-14-01630-f003:**
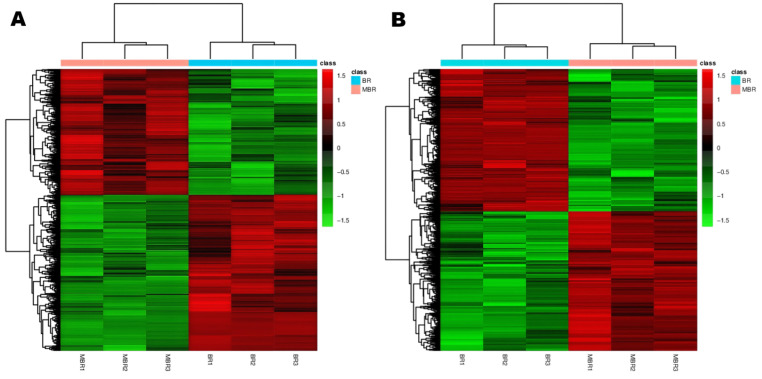
Differential ion cluster analysis graph of BR:MBR, BR:WBR, and HTP-WBR:WBR. Each row in the figure represents a differential ion, and each column represents a sample. Different colors indicate different intensities, and colors range from green to red, indicating strength from low to high. (**A**): positive ions; (**B**): negative ions.

**Figure 4 foods-14-01630-f004:**
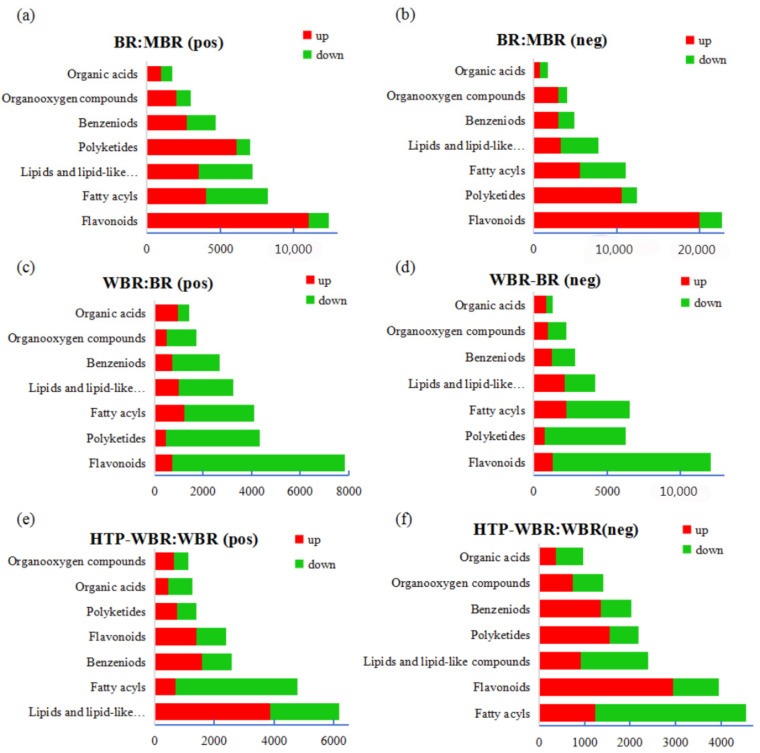
Major classifications of differential metabolites (the top seven classifications) enriched in the positive (pos) and negative (neg) ion modes: (**a**,**b**) differential numbers of metabolites in BR:MBR under positive and negative ion conditions; (**c**,**d**) differential numbers of metabolites in BR:WBR under positive and negative ion conditions; and (**e**,**f**) differential numbers of metabolites in WBR:HTP-WBR under positive and negative ion conditions.

**Figure 5 foods-14-01630-f005:**
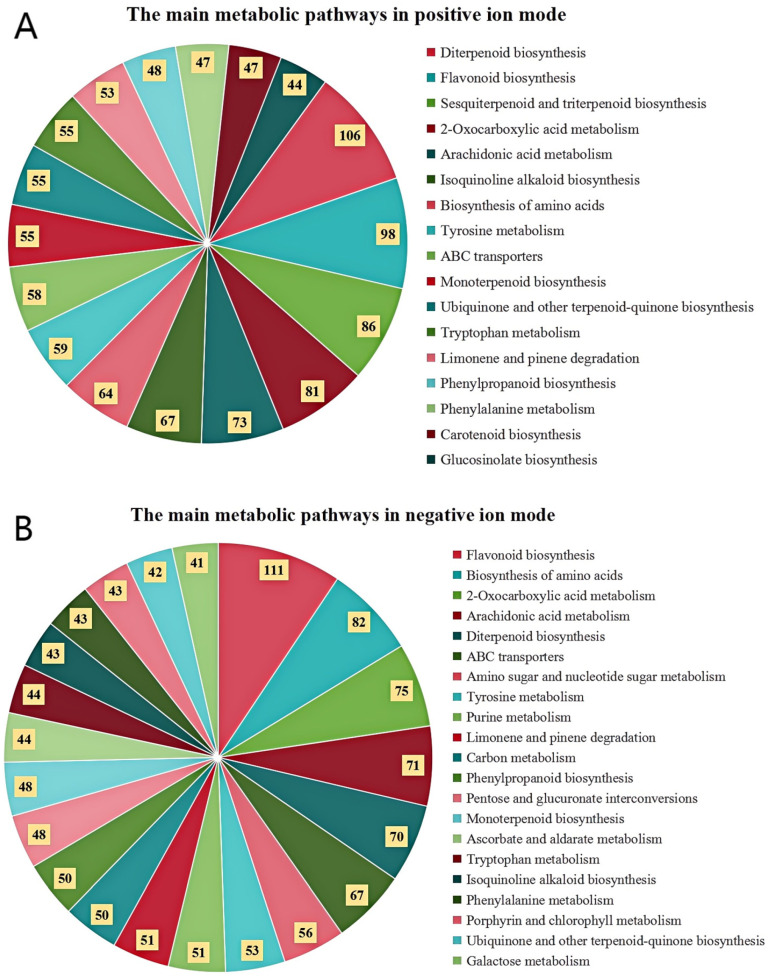
The primary metabolic pathways based on KEGG database.

**Figure 6 foods-14-01630-f006:**
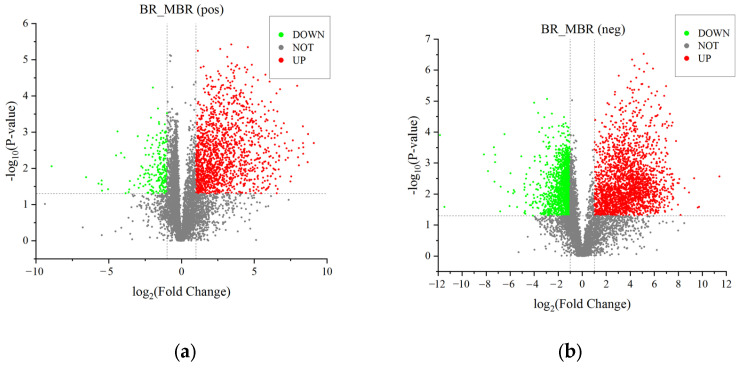
Volcano plots of the differential metabolites: (**a**,**b**) volcano plots of BR and MBR; (**c**,**d**) volcano plots of BR and WBR; and (**e**,**f**) volcano plots of WBR and HTP-WBR. Red dots: Significantly upregulated points (points satisfying *p* < 0.05 and fold change ≤1.5). Green dots: Significantly downregulated points (points satisfying *p* < 0.05 and fold change ≤0.67). Gray dots: Non-significant points (points that do not meet the above criteria).

**Table 1 foods-14-01630-t001:** Major parameters of Progenesis QI.

Detection Mode	Adduct Ion	Peak Alignment	Peak Extraction Parameters	Database	Mass Deviation
positive ions	[M+K]^+^, [M+NH4]^+^, [M+Na]^+^, [M+H]^+^, [M+H−H_2_O]^−^	The system automatically selects the most suitable sample from the QC samples as the reference sample.	automatic	KEGG, HMDB, LipidMaps	10^−6^
negative ions	[M+FA−H]^−^, [M−H]^−^, [M−H_2_O−H]^−^, [M+Cl]^−^	ditto	ditto	ditto	ditto

**Table 2 foods-14-01630-t002:** All ion number and identification results.

Mode ^(1)^	Total Ion Number	RSD ≤ 30% Ion Number ^(2)^	MS1 ^(3)^	MS2 ^(4)^
positive ions	8140	6988	5011	3026
negative ions	9940	7099	4923	3452

^(1)^ positive ions refer to the addition of H+, NH4+, Na+, and K+, etc., when the material is ionized at the ion source; negative ions refer to the addition of -H, +Cl, and +OAc ions when the ion is ionized at the ion source; total ion number refers to the total number of ions obtained after peaking by employing Progenesis QI after the MS data are removed. ^(2)^ RSD ≤ 30% ion number: the number of ions satisfying RSD ≤ 30%. The ions that met this condition were used for subsequent statistical analysis. ^(3)^ MS1: the first identification number represents the ions identified after processing the initial data through a database search; the initial data refers to the mass spectrum output of metabolites, captured after the ion source introduces adduct ions such as H+/NH4+/COOH-. ^(4)^ MS2: the secondary identification number represents the quantity of ions identified following a database search for theoretical secondary fragments. Secondary data correspond to the fragment ion spectra of metabolites, produced by breaking down the first data in the mass spectrometer.

**Table 3 foods-14-01630-t003:** Number of differential ions in different BR forms.

Mode	Groups	Differential Ion Number	Up (MS1)	Down (MS1)	Up (MS2)	Down (MS1)
positive	BR:MBR	3075	1698	1377	1420	2319
	BR:WBR	1700	978	722	793	1324
	WBR:HTP-WBR	1624	798	826	1251	766
negative	BR:MBR	4434	2245	2189	2191	3076
	BR:WBR	2324	1120	1204	1151	1666
	WBR:HTP-WBR	1426	820	606	740	1077

**Table 4 foods-14-01630-t004:** Essential metabolites involved in differential metabolic pathways.

Mode	Pathway	Pathway ID	Metabolites	BR:MBR		BR:WBR		WBR:HTP-WBR	
				Up	Down	Up	Down	Up	Down
positive	Flavonoid biosynthesis	map00941map00944	Phenylpropanoids and polyketides	214	92	245	85	189	141
	Amino acid biosynthesis	map01230	Organic acids and derivatives, lipids and lipid-like molecules, and fatty acyls	123	69	104	88	62	130
	2-Oxocarboxylic acid metabolism	map01210	Organic acids and derivatives, lipids and lipid-like molecules, and fatty acyls	99	41	75	65	34	106
	Arachidonic acid metabolism	map00590	Organic acids and derivatives, lipids and lipid-like molecules, and fatty acyls	547	183	529	201	97	633
	Terpenoid biosynthesis	map00904map00902map00909	Prenol lipids, lipids and lipid-like molecules, benzenoids, and fatty acyls	529	330	501	398	441	442
negative	Flavonoid biosynthesis	map00941map00944	Phenylpropanoids and polyketides	480	52	491	51	138	404
	Amino acid biosynthesis	map01230	Organic acids and derivatives, lipids and lipid-like molecules, and fatty acyls	207	130	175	162	126	77
	2-Oxocarboxylic acid metabolism	map01210	Organic acids and derivatives, lipids and lipid-like molecules, and fatty acyls	131	75	115	90	42	86
	Arachidonic acid metabolism	map00590	Organic acids and derivatives, lipids and lipid-like molecules, and fatty acyls	901	170	894	177	183	877
	Terpenoid biosynthesis	map00904map00902map00909	Prenol lipids, lipids and lipid-like molecules, benzenoids, and fatty acyls	116	167	165	118	97	186

## Data Availability

The data have been uploaded to the Zenodo database, and the DOI link is: 10.5281/zenodo.15334777.

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
