# Peer review of "Changes in Bioactive Constituents in Black Rice Metabolites Under Different Processing Treatments"

_foods, 2025, doi:10.3390/foods14091630_

Round 1

Reviewer 1 Report

Comments and Suggestions for Authors

Foods-3470193.

General

The manuscript deals with the study of black rice (BR), BR after milling (MBR), wet germinated black rice (WGBR), and high-temperature-and high-pressure-treated WBR) to perform an untargeted metabolomic essay, using liquid chromatography-mass spectrometry. The study involves the determination of many compounds and their comparison over the different processing steps. The manuscript is clear and concise. However, metabolomic variations in the title could not correctly reflect the cause of the changes, which are not derived from natural or normal evolution but caused by different treatments. Maybe Metabolite changes could be more realistic. Also, the use of the text for expressions such as biosynthesis or metabolism is not precise. The manuscript should state clearly that provoked treatments induced changes. Only in the case of wet germination could it possibly be adequately used. Alternatively, they could justify their point of view more vigorously.  Additionally, the legend size of several figures should be increased to make them easily legible, particularly Fig. 2.

Specific

L108. How the 30% criterium was adopted?

L121. How were these values chosen?  

L134. Supposedly, the criteria were explained previously.

Figure 2. The legend size should be increased to make them legible.

Figure 5. This figure should be explained in more detail, specifically regarding how the significant differences were evaluated.

L 323. The metabolic characteristics…. Maybe the metabolic or the metabolite profiles…. 

Reviewer 2 Report

Comments and Suggestions for Authors

"The manuscript by Hong et al. deals with a metabolomic characterization of black rice (BR), milled black rice, wet-germinated black rice, and high-temperature- and high-pressure-treated wet-germinated black rice (WBR). While the subject matter is of interest and aligns with the journal's scope, several methodological and analytical deficiencies warrant a request for substantial revision prior to potential re-submission.

Methodological Concerns:

  • The manuscript lacks detailed information regarding sample preparation, specifically the extraction procedures employed (solvents, Temperature, etc). Furthermore, validation of the multivariate analysis is absent, compromising the robustness of the data interpretation.

Analytical Deficiencies:

  • The methodology for mass assignment is inadequately described. Relying solely on retention time without supporting fragmentation data raises concerns regarding the reliability of metabolite identification.
  • The absence of metabolite quantification is a significant omission, particularly in the context of food analysis, where quantitative data are crucial for assessing compositional changes.
  • In the statistical analysis, specifically the Principal Component Analysis (PCA), the corresponding loadings are not provided, hindering a thorough evaluation of the variables contributing to the observed variance.

In conclusion, the authors are encouraged to address these critical issues through comprehensive revisions. Re-submission is contingent upon the satisfactory resolution of these methodological and analytical shortcomings."

Reviewer 3 Report

Comments and Suggestions for Authors

The introduction well written and clearly introduced the background information and the objectives of this study. Minor revision is needed to improve the justification and clarity.

  1. What is the unique property of black rice compared to other rice varieties.
  2. It is better to provide more detail on the treatment methods like germination, heat-moisture treatment, and the importance to compare those methods.
  3. Research gap need further clarified.

The materials and methods section is well written, minor revision is needed to improve the clarity

  1. Keep the units consistent.
  2. Check typos thoroughly.
  3. It is better to move the 2.1 sample treatment part to the 2.3 section and involve more details for each treatment.
  4. Line 131-132: not clear.
  5. Line 172: any citation for this self-developed software if published or briefly describe its key features.
  6. Section 2.3.7: It would be better to list the specific non-parametric or parametric tests, such as Student t-test or Man-Whitney U test, used in the univariate analysis. Same for the cluster analysis.

Result

  1. Section 3.1: More interpretation of the data needed. For example, it would be better to involve more info on the significance of the identified ions. Providing more info on their metabolites or potential roles would help understand the practical implications of the findings.
  2. Section 3.2: it would be better to involve more info on the significance of these pathway. How these metabolites contribute to health benefits. It would be better to explain the methods used for classification and functional analysis.
  3. Section 3.3: more details on the implications of the metabolic cluster findings could help. And explain how this separation influence the understanding of the back rice properties. Explain why processing methods cause the differences.
  4. Section 3.4: It would be better to clarify the relationships between metabolites and functionalities and the significance of these pathways on health benefits or potential applications.

Discussion

  1. It would be better to compare with literature findings.
  2. It would be better to provide more mechanistic understanding of the findings. For example, discussing how those processing/treatment method/conditions impact the functional properties/health benefits of the identified metabolites.
  3. Summarize the key findings with implications

Round 2

Reviewer 1 Report

Comments and Suggestions for Authors

Most of the suggestions were addressed. No further comments.

Author Response

Thank you for your help with this article!

Reviewer 3 Report

Comments and Suggestions for Authors

The manuscript has improved a lot after the revision. However, one comment may not have been clearly understood. In Section 2.3.1, the sentence "Black Pearl rice was planted in Weiguo Township, Wuchang City, Heilongjiang Province (BR)" is about the source of the material, so it would be better to move it to Section 2.1 and change the title to "Materials and Instruments" or similar.  Also, I suggest changing the title of Section 2.3.1 from "Rice samples" to "Different treatments of rice samples" or something similar, to better match the content of that section. This will make the experimental design much clear and also reflect the title of this study. 

Author Response

Comment:In Section 2.3.1, the sentence "Black Pearl rice was planted in Weiguo Township, Wuchang City, Heilongjiang Province (BR)" is about the source of the material, so it would be better to move it to Section 2.1 and change the title to "Materials and Instruments" or similar. Also, I suggest changing the title of Section 2.3.1 from "Rice samples" to "Different treatments of rice samples" or something similar, to better match the content of that section. This will make the experimental design much clear and also reflect the title of this study. 

Response:Thank you for pointing this out.

Section 2.1 has been revised to "Materials and Instruments", divided into "2.1.1 Instruments" and "2.1.2 Materials". The statement "Black Pearl rice was planted in Weiguo Township, Wuchang City, Heilongjiang Province (BR)" has been placed under the "2.1.2 Materials" subsection. Additionally, section 2.2.1 has been modified from "Rice samples" to "Different treatments of BR samples".
